# Short versus long cephalomedullary nails for intertrochanteric femur fractures: A meta-analysis of randomized controlled trials

Shengquan Zhang[1], Qiaofeng Guo[2], Kai Huang[2]*, Haiqun Zhu[2]*

**1** Department of Trauma, Hangzhou Fuyang Hospital of TCM Orthopedics and Traumatology, Hangzhou, Zhejiang, China, **2** Department of Orthopedics, Tongde Hospital of Zhejiang Province, Hangzhou, Zhejiang, China

* zjhzhuangkai@163.com (KH); zhq13989829084@163.com (HZ)

## Abstract

### Purpose

We aim to evaluate the efficacy and safety of short cephalomedullary nails(CMN) versus long CMN in patients with intertrochanteric femur fractures(IFFs).

### Methods

The PubMed, Web of Science, and Embase databases were searched for relevant publications until July 2024. All randomized controlled studies evaluating the efficacy and safety of short CMN versus long CMN in patients with IFFs were included. We estimated the pooled risk ratio (RR) with 95% confidence intervals (CIs) for binary outcomes, and the mean difference (MD) for continuous outcomes.

### Results

A total of 7 studies with 658 patients were included in this analysis. There was no significant difference between the short CMN group and the long CMN group in Harris hip score, mortality within 1-year, overall complication rates, or reoperation rates. However, durations of surgery were significantly lower in the short CMN group compared to the long CMN group (MD: −21.83 minutes, 95% CI: −27.54 minutes, −16.13 minutes), along with significantly lower intraoperative blood loss (MD: −136.70 mL, 95% CI: −139.06 mL, −134.34 mL) and tip-apex distance (MD: −0.47 cm, 95% CI: −0.63 cm, −0.31 cm). There was also no significant difference in peri-implant fracture or lengths of hospital stays.

**Data availability statement:** All relevant data are within the paper and its Supporting Information files.

**Funding:** This study was funded by Zhejiang Clinovation Pride (Clinical Innovation Team for Traumatic Osteomyelitis) (NO. CXTD202501009).

**Competing interests:** The authors have declared that no competing interests exist.

## Conclusions

Short CMN are associated with shorter duration of surgery, reduced tip-apex distance, and lower intraoperative blood loss compared to long CMN for the fixation of IFFs. However, there were no significant differences in functional outcomes, overall complication rates, reoperation rates, mortality within one year, peri-implant fracture, or lengths of hospital stays.

## 1. Introduction

Intertrochanteric femur fractures (IFFs) are a common type of proximal femur fracture, particularly prevalent among the elderly population due to factors such as osteoporosis and decreased bone density [1]. With the global increase in life expectancy, the incidence of IFFs has risen significantly, placing a substantial burden on healthcare systems worldwide [2]. These fractures are associated with high morbidity and mortality rates, particularly when surgical intervention is delayed [3]. Therefore, the timely and effective surgical fixation of IFFs is of paramount importance in improving patient outcomes.

For the surgical fixation of IFFs, two primary methods have been utilized: cephalomedullary nails (CMN) and dynamic hip screws (DHS) [4,5]. While both techniques are considered effective for stabilizing stable IFFs, CMN have shown superior performance in the fixation of unstable fractures [6]. The advantage of CMN lies in their ability to provide better biomechanical stability, particularly in complex fracture patterns, reducing the risk of implant failure and complications such as screw cut-out [7]. However, several recent reviews suggest that the differences between CMN and DHS are small, particularly in unstable fractures such as AO type A2 fractures, although not including reverse obliquity fractures [8,9]. Despite this, CMN are still increasingly preferred over DHS for the treatment of unstable intertrochanteric fractures, partly due to their ability to provide superior stability in certain fracture patterns [10]. This change in practice might also be influenced by factors such as surgeon preference, hospital protocols, and regional variations [11].

Within the category of CMN, both short and long versions are available, each with its own set of benefits and limitations [12]. Short CMN are associated with reduced surgical time, less intraoperative blood loss, and decreased need for blood transfusions [13]. On the other hand, long CMN offer the potential benefit of reducing stress concentration at the distal end of the implant, thereby lowering the risk of secondary fractures of the femoral shaft [14]. Despite these advantages, there remains ongoing debate regarding the superiority of short versus long CMN in terms of overall efficacy and safety for the fixation of IFFs. The existing literature provides conflicting evidence [15–18].

Given the lack of consensus and the varying results reported in previous studies, this meta-analysis aims to systematically evaluate the efficacy and safety comparing short versus long CMN for the fixation of IFFs.

## 2. Materials and methods

The meta-analysis was performed according to the Preferred Reporting Items for a Systematic Review and Meta-analysis (PRISMA) 2020 guidelines [19]. We registered the protocol for this meta-analysis with PROSPERO under the identifier CRD42024616788.

### 2.1. Search strategy

A comprehensive literature search was conducted in PubMed, Web of Science, and Embase databases up to July 2024. The search strategy utilized the following keywords: ("Fracture Fixation, Intramedullary" OR "Intramedullary Nailing" OR "Intramedullary Nail" OR "Cephalomedullary Nail" OR "Cephalomedullary Nailing") AND ("Intertrochanteric Femur Fractur*" OR "Intertrochanteric Fractur*" OR "Hip Fractur*" OR "Proximal Femur Fractur*" OR "Pertrochanteral Fractur*" OR "Pertrochanteric Fractur*"). Detailed search parameters are available in Supplementary Table 1 in S3 File. Additionally, the reference lists of selected articles were manually reviewed to identify further relevant studies.

### 2.2. Inclusion and exclusion criteria

Studies were included based on the following PICOS criteria: Population (P): patients diagnosed with IFFs; Intervention (I): short CMN (length ranging from 170 to 200 mm); Comparison (C): long CMN (length ranging from 240 to 460 mm); Outcomes (O): extractable data for at least one of the following outcomes: durations of surgery, tip-apex distance, intraoperative blood loss, mortality within 1-year, reoperation rates, lengths of hospital stay, Harris hip score, and overall complication rates; Study Design (S): randomized controlled trials with a sample size of over 10 patients.

Exclusion criteria included studies conducted on animals and articles not meeting the inclusion criteria, such as reviews, case reports, conference abstracts, meta-analyses, letters to editors, case-control studies, cohort studies, and cross-sectional studies. Additionally, studies published in languages other than English were excluded due to accessibility issues for readers and concerns regarding the consistency of study quality across different languages.

### 2.3. Quality assessment

The quality of the included studies was assessed using the Cochrane Risk of Bias Tool for randomized trials [20]. This tool evaluates potential biases in several domains, including selection bias, performance bias, detection bias, attrition bias, and reporting bias. Each domain was carefully examined to ensure a thorough and objective assessment.

Two independent reviewers conducted the quality assessment, with any disagreements resolved through discussion or consultation with a third reviewer. The findings from the risk of bias assessment were used to gauge the overall quality of evidence and to inform subsequent analyses in this meta-analysis.

### 2.4. Data extraction

Data extraction from all relevant articles was independently conducted by two researchers. Collected information included author names, publication year, study characteristics (e.g., country, study design, measured outcomes, and comparison), as well as patient demographics such as age, gender ratio (female/male), and total patient count. Any discrepancies between the researchers were resolved through discussion until consensus was achieved, ensuring consistency and reliability in the data extraction process. When data were not reported or were incomplete, we first attempted to contact the corresponding authors to request additional information. In particular, we made several attempts to contact the authors of Shannon et al. [18], but unfortunately, no response was received. If no response was received, we performed an analysis of the available data. In cases of unresolved missing data, we performed a complete case analysis using only the available data, which may potentially introduce some bias into our meta-analysis.

## 2.5. Outcome measures

The primary outcome measures were functional outcomes, specifically the Harris Hip Score, and mortality within 1 year. Secondary outcomes included complication rates(any adverse events occurring during or after surgery), reoperation rates, the duration of surgery (time from the initial incision to final wound closure), tip-apex distance, intraoperative blood loss, and length of hospital stay.

## 2.6. Statistical analysis

For continuous outcomes, such as duration of surgery and tip-apex distance, the pooled effect estimates were calculated using the mean difference (MD), each accompanied by a 95% confidence interval (CI)[21]. For binary outcomes, such as 1-year mortality rates and reoperation rates, risk ratios (RR) were used to determine the effect estimates, also with 95% CIs [22]. Heterogeneity within and between studies was assessed using the Cochrane Q test and $I^2$ statistic [23]. If significant heterogeneity was detected ($I^2 \geq 50\%$), a random-effects model was applied; otherwise, a fixed-effects model was used [23]. For outcomes with significant heterogeneity, leave-one-out sensitivity analyses were performed to identify the sources of heterogeneity and assess the robustness of the results [24].

Publication bias for the duration of surgery, was evaluated using funnel plots and Egger's test [25]. A *P*-value of less than 0.05 was considered statistically significant. All statistical analyses were conducted using R software version 4.3.1.

## 3.1. Results

### 3.1. Literature search and study selection

A comprehensive search of three databases initially identified 4,224 studies. After removing 2,092 duplicates, 2,104 studies were excluded during the initial screening for not meeting the inclusion criteria. Full-text reviews were conducted on the remaining 25 articles, leading to the exclusion of an additional 18 studies: 6 did not compare short versus long CMN, 10 were not randomized controlled trials, and 2 were non-English publications. Ultimately, 7 randomized controlled trials were selected for inclusion in the meta-analysis to evaluate the efficacy and safety of short versus long CMN for IFFs [5, 12, 15–18, 26]. The PRISMA flow diagram in Fig 1 illustrates the selection process.

### 3.2. Study description and quality assessment

This meta-analysis included seven randomized controlled trials, encompassing a total of 658 patients, with individual study sample sizes ranging from 33 to 168 participants. The complications reported in the studies were as follows: Dragosloveanu et al. (2022) identified screw cut-out and local postoperative hematoma; Garín et al. (2022) noted mechanical issues such as blade lateral migration, cut-through, malrotation, pin fatigue/fracture, and non-union, along with infections and other general complications; Galanopoulos et al. (2018) reported the z-effect phenomenon and periprosthetic fractures; Okcu et al. (2013) observed superficial and deep surgical site infections; and Shannon et al. (2019) mainly reported screw cut-out and deep surgical site infection. The characteristics of the patients and the technical aspects of the interventions in each study are succinctly summarized in Table 1.

The risk of bias for each study was assessed using the Cochrane Risk of Bias Tool, with the results presented in Fig 2. A high risk of bias was identified in the domain of blinding of participants and personnel (performance bias) in six out of the seven studies. This high risk primarily reflects the challenges in blinding surgical interventions such as short versus long CMN, where both the surgeon and patient are often aware of the treatment being administered. Despite this, the overall quality of the included studies was deemed acceptable for drawing conclusions regarding the efficacy and safety of the interventions.

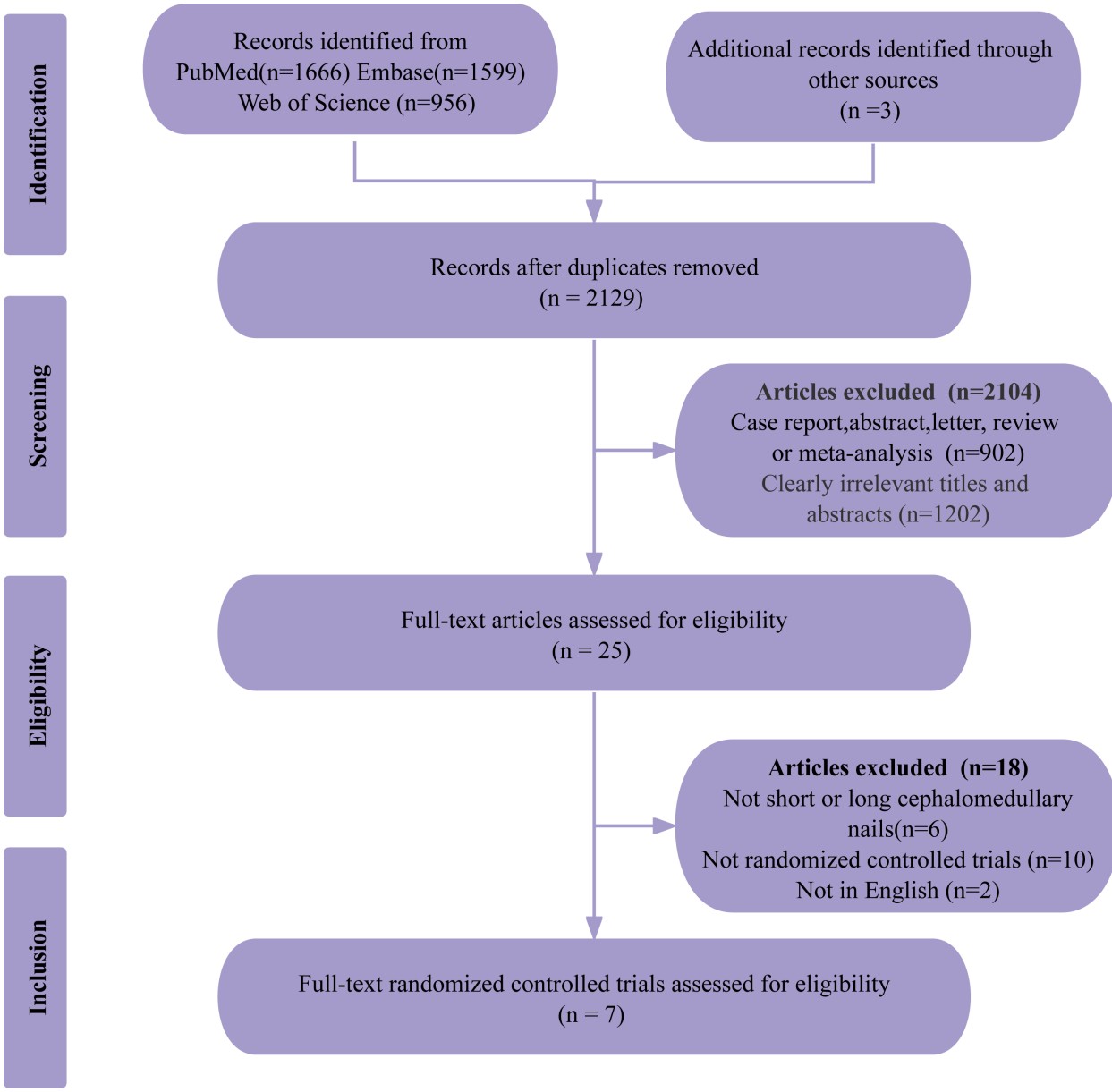

**Fig 1. PRISMA flow diagram illustrating the study selection process.**

### 3.3. Harris hip score

For Harris hip score, 4 studies were analyzed and random effect models were employed due to the significant heterogeneity ($I^2 = 88\%, P < 0.01$). The Harris hip score have no significant difference between short CMN group compared to the long CMN group (MD: 2.07, 95% CI: -1.98, 6.12) (Fig 3).

Leave-one-out sensitivity analysis indicated that no clear source of heterogeneity could be identified. Excluding Okcu et al. 2013, the results became inconsistent with previous conclusions, suggesting that these findings require cautious interpretation (Supplementary Figure 1 in S3 File).

**Table 1. The study characteristics of the included studies.**

| Author | Year | Country | Study design | Outcome | Comparison | Mean age±SD or range | AO/OTA Fracture | Male/ Female | Number of patients |
|---|---|---|---|---|---|---|---|---|---|
| Dragosloveanu et al. | 2022 | Romania | Randomized controlled trial | (1)(2)(4)(5) (6)(7)(8)(9) | Short CMN | 78.65±4.84 | 31-A2(80.7%), 31-A3(19.3%) | 14/12 | 26 |
| | | | | | Long CMN | 78.07±5.03 | 31-A2(77.7%), 31-A3(22.3%) | 13/14 | 27 |
| Garín et al. | 2022 | Spain | Randomized controlled trial | (1)(2)(3)(5) (6)(9) | Short CMN | 85.0±7.1 | 31-A2(82%), 31-A3(18%) | 19/71 | 90 |
| | | | | | Long CMN | 84.9±7.3 | 31-A2(85%), 31-A3(15%) | 20/47 | 67 |
| Galanopoulos et al. | 2018 | Greece | Randomized controlled trial | (1)(4)(5)(9) | Short CMN | 81 (74–92) | 31-A2,31-A3 | NA | 25 |
| | | | | | Long CMN | 79 (74–93) | 31-A2,31-A3 | NA | 25 |
| Okcu et al. | 2013 | Turkey | Randomized controlled trial | (1)(2)(3)(5) (8)(9) | Short CMN | 78 (67–95) | 31-A3(100%) | 4/11 | 15 |
| | | | | | Long CMN | 81 (73–89) | 31-A3(100%) | 4/14 | 18 |
| Sahu et al. | 2020 | India | Randomized controlled trial | (1)(3)(4)(5) (6)(7)(8) | Short CMN | 75.2±8.3 | 31-A1(33.3%), 31-A2(53.3%), 31-A3(13.3%) | 15/30 | 45 |
| | | | | | Long CMN | 77.3±9.8 | 31-A1(28.6%) 31-A2(50%) 31-A3(21.4%) | 15/27 | 42 |
| Sellan et al. | 2019 | Canada | Randomized controlled trial | (1)(3)(4)(6) | Short CMN | Overall, 79 (56–97) | 31-A1(15.5%),31-A2 and 31-A3(84.5%) | Over-all,35/75 | 71 |
| | | | | | Long CMN | | 31-A1(15.4%),31-A2 and 31-A3(84.6%) | | 39 |
| Shannon et al. | 2019 | USA | Randomized controlled trial | (1)(2)(4)(5) (7)(8)(9) | Short CMN | (79–84) | 31-A1(16.2%),31-A2(76.3%),31-A3(7.5%) | NA | 80 |
| | | | | | Long CMN | (76–82) | 31-A1(13.6%),31-A2(76.1%),31-A3(10.2%) | NA | 88 |

NA not available; CMN cephalomedullary nails; AO Arbeitsgemeinschaft für Osteosynthesefragen; OTA orthopaedic trauma association; (1) Duration of surgery; (2) Tip-apex distance; (3) Mortality within 1 year; (4) Peri-implant fracture; (5) Reoperation rates; (6) Lengths of hospital stays; (7) Intraoperative blood loss; (8) Harris hip score; (9) Overall complication rates.

### 3.4. Mortality within 1 year

For mortality within 1 year, 4 studies were analyzed using fixed effect models due to the absence of significant heterogeneity ($I^2=3\%$ and $P=0.36$). The rates of mortality within 1 year have no significant difference between short CMN group compared to the long CMN group (RR: 0.78, 95% CI: 0.46, 1.31) (Fig 4).

### 3.5. Overall complication rates

For overall complication rates, 5 studies were analyzed and fixed effect models were employed due to the low heterogeneity ($I^2=0\%$,$P=0.99$). The overall complication rates have no significant difference between short CMN group compared to the long CMN group (RR:1.04, 95% CI: 0.71, 1.53) (Fig 5).

### 3.6. Reoperation rates

For reoperation rates, 6 studies were analyzed and fixed effect models were employed due to the absence of significant heterogeneity ($I^2=0\%$,$P=0.89$). The rates of reoperation have no significant difference between short CMN group compared to the long CMN group (RR: 0.59, 95% CI: 0.28, 1.22)(Fig 6).

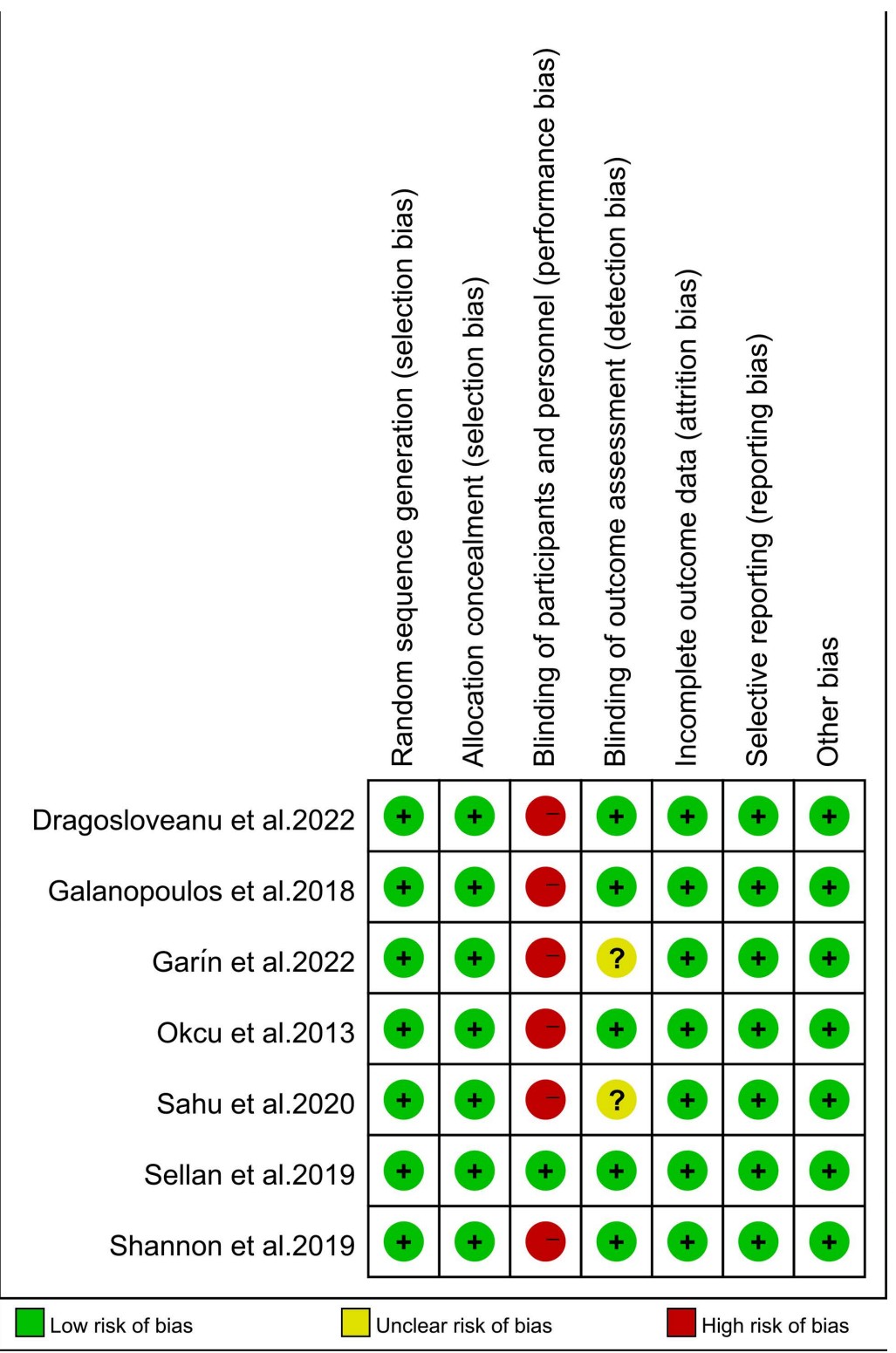

**Fig 2. Assessment of study quality using the Cochrane Risk of Bias Tool: a color-coded evaluation of each domain (low risk: green, unclear risk: yellow, high risk: red).**

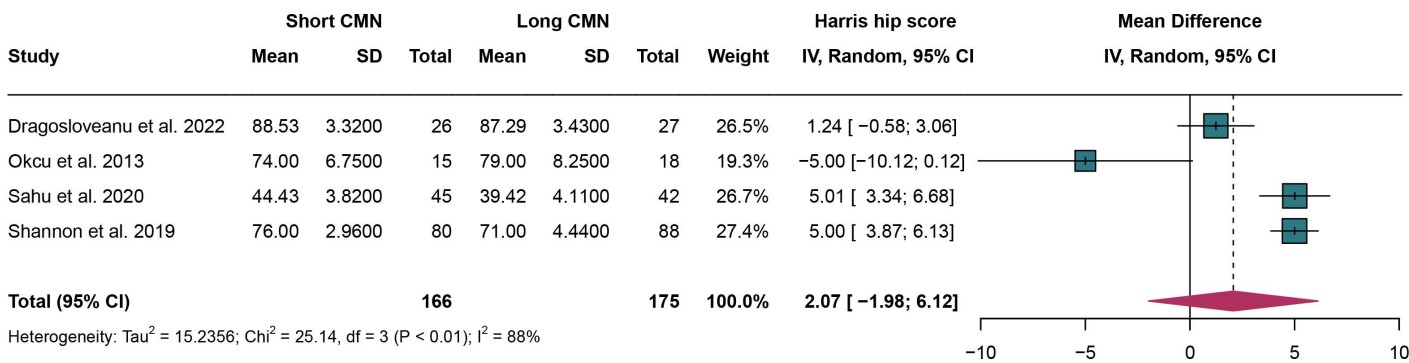

**Fig 3. Forest plot comparing Harris Hip Score between short and long cephalomedullary nails groups.**

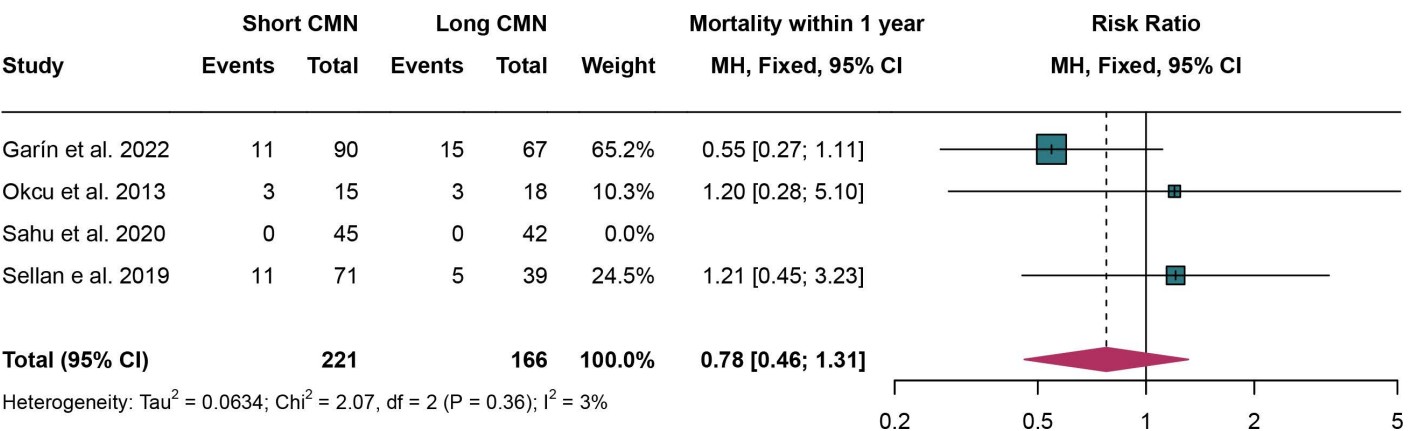

**Fig 4. Forest plot comparing mortality within 1 year between short and long cephalomedullary nails groups.**

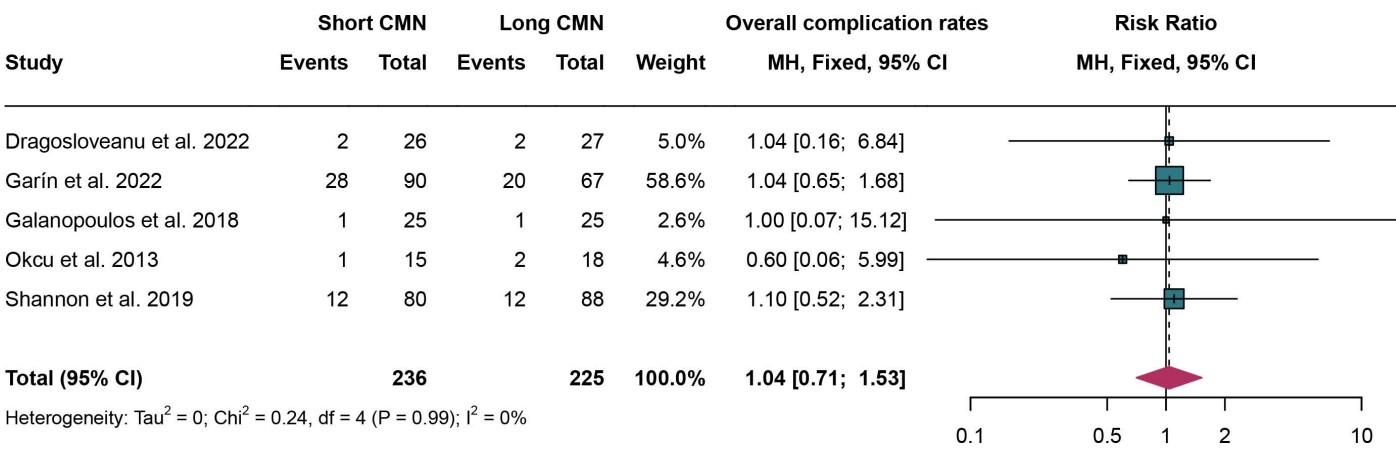

**Fig 5. Forest plot comparing overall complication rates between short and long cephalomedullary nails groups.**

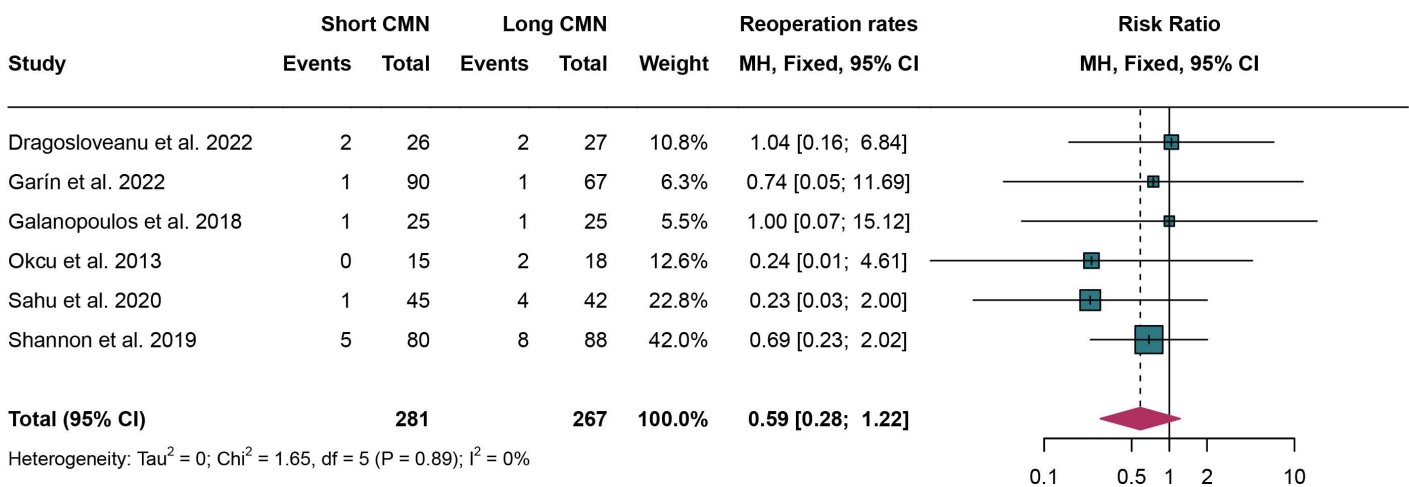

**Fig 6. Forest plot comparing reoperation rates between short and long cephalomedullary nails groups.**

### 3.7. Duration of surgery

For duration of surgery, a total of 7 studies were analyzed and random effect models were applied as significant heterogeneity($I^2 = 91\%$ and $P < 0.01$). The durations of surgery were significantly lower in the short CMN group compared to the long CMN group (MD: -21.83 minutes, 95% CI: -27.54 minutes, -16.13 minutes) (Fig 7).

Leave-one-out sensitivity analysis indicated that no clear source of heterogeneity could be identified, as the $I^2$ statistic remained above 50% even after omitting studies individually. However, the overall results remained stable despite the persistent heterogeneity (Supplementary Figure 2 in S3 File). Additionally, funnel plot analysis and Egger's test showed no significant publication bias ($P = 0.07$) (Fig 8). However, it is worth noting that approximately 50% of the included studies fell outside the funnel plot, which may raise concerns about the potential influence on the generalizability of these results.

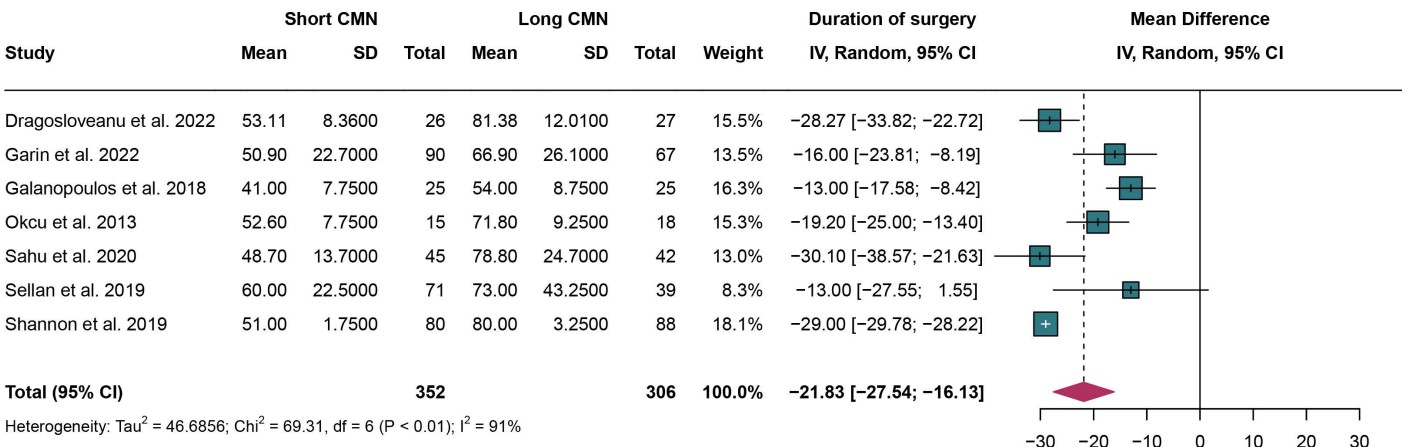

**Fig 7. Forest plot comparing duration of surgery between short and long cephalomedullary nails groups.**

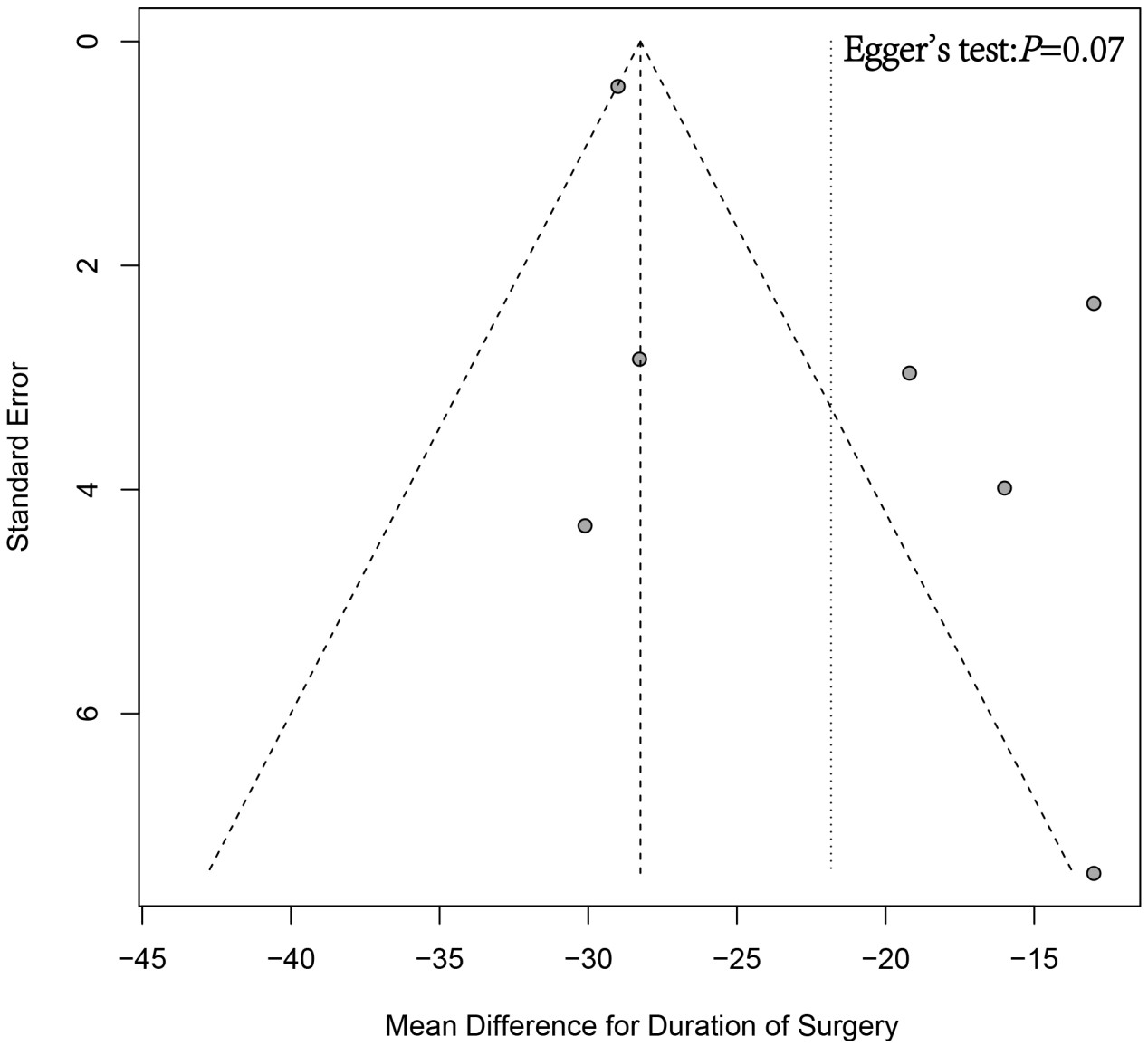

**Fig 8. Funnel plot and Egger's test for publication bias analysis in the duration of surgery between short and long cephalomedullary nails groups.**

### 3.8. Intraoperative blood loss

For intraoperative blood loss, 3 studies were analyzed and fixed effect models were employed due to the low heterogeneity ($I^2=0\%, P=0.47$). The short CMN group have a significantly lower intraoperative blood loss compared to the long CMN group (MD: -136.70 mL, 95% CI: -139.06 mL, -134.34 mL)([Fig 9]).

### 3.9. Tip-apex distance

For tip-apex distance, a total of 4 studies were analyzed and fixed effect models were applied as no significant heterogeneity($I^2=44\%$ and $P=0.15$). Short CMN showed a weighted mean TAD of 18.3 mm, while long CMN demonstrated a

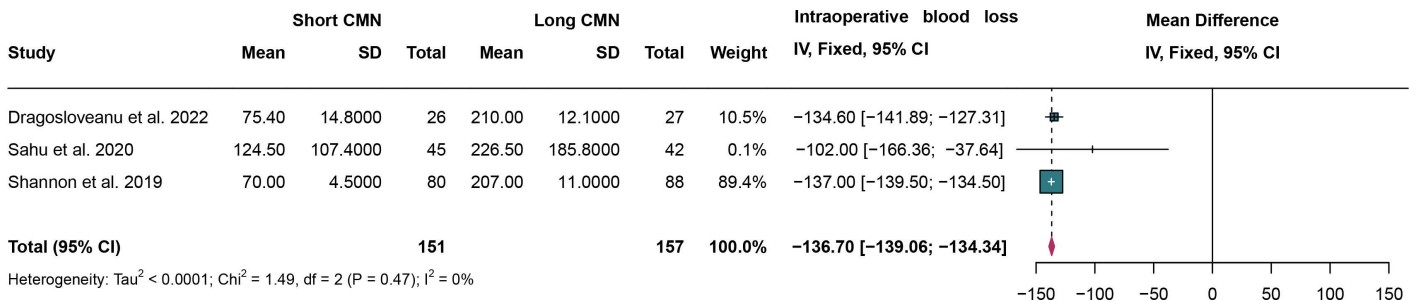

**Fig 9. Forest plot comparing intraoperative blood loss between short and long cephalomedullary nails groups.**

weighted mean TAD of 19.0 mm. The tip-apex distance was significantly lower in the short CMN group compared to the long CMN group (MD: -0.47 cm, 95% CI: -0.63 cm, -0.31 cm) (Fig 10).

### 3.10. Peri-implant fracture

For peri-implant fracture, 5 studies were analyzed and fixed effect models were employed due to the absence of significant heterogeneity ($I^2 = 0\%$, $P = 0.56$). The rates of peri-implant fracture have no significant difference between short CMN group compared to the long CMN group (RR: 1.05, 95% CI: 0.39, 2.82) (Fig 11).

### 3.11. Lengths of hospital stays

For lengths of hospital stays, 4 studies were analyzed and random effect models were employed due to the significant heterogeneity ($I^2 = 51\%$, $P = 0.11$). The lengths of hospital stays have no significant difference between short CMN group compared to the long CMN group (MD: -0.29 days, 95% CI: -1.08 days, 0.50 days)(Fig 12).

Leave-one-out sensitivity analysis revealed that after omitting the study by Sahu et al.2020, the $I^2$ statistic decreased to 39%, suggesting that this study might be a potential source of the observed heterogeneity. However, the overall results remained stable despite this reduction in heterogeneity (Supplementary Figure 3 in S3 File).

## 4. Discussion

In the treatment of IFFs, intramedullary nails are a widely accepted and effective option, with both long and short CMN offering distinct advantages [27]. Our meta-analysis demonstrated that short CMN are associated with significantly shorter durations of surgery, lower tip-apex distance, and reduced intraoperative blood loss compared to long CMN in the fixation of IFFs. However, these differences were statistically significant but small in absolute terms. These findings may be attributed to the technical differences between the two types of nails. Short CMN are quicker to insert due to their reduced length, which requires less reaming of the femoral canal and simpler distal locking procedures [15,16,18]. The shorter length of short CMN minimizes the distance the implant must traverse, potentially simplifying the alignment process during surgery and allowing for more precise placement of the lag screw within the femoral head [18,26]. This precision may result in a reduced tip-apex distance, as the surgeon may have greater control over the screw trajectory and final positioning when dealing with shorter implants [12,15]. Furthermore, the reduced length of short nails may decrease procedural complexity in aligning the distal nail, allowing surgeons to focus more on achieving optimal screw placement, which is a crucial factor in minimizing complications such as screw cut-out [18]. Furthermore, the less invasive nature of the procedure with short CMN results in lower intraoperative blood loss, likely due to decreased soft tissue and bone disruption during insertion [15,16,18].

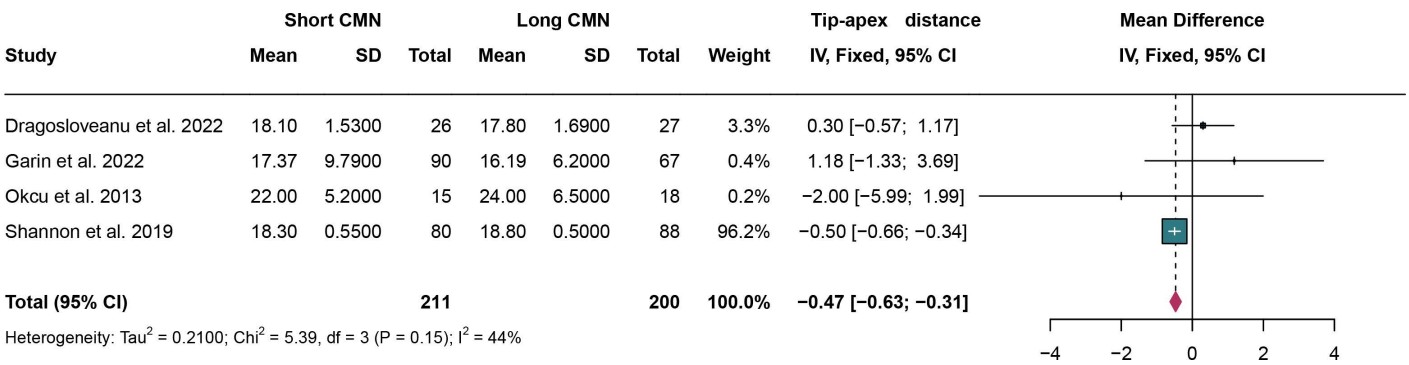

**Fig 10. Forest plot comparing tip-apex distance between short and long cephalomedullary nails groups.**

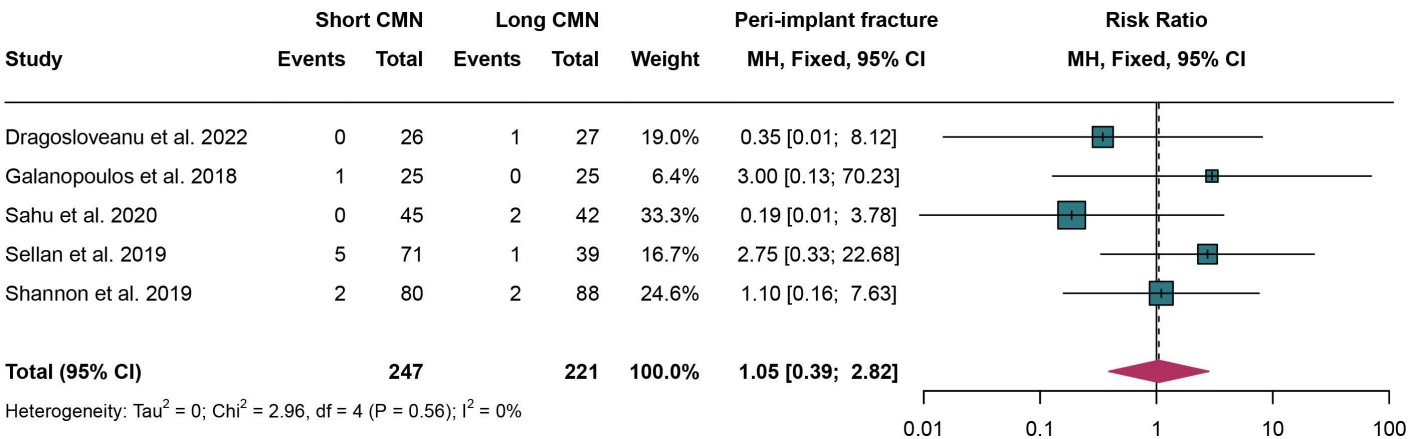

**Fig 11. Forest plot comparing peri-implant fracture rates between short and long cephalomedullary nails groups.**

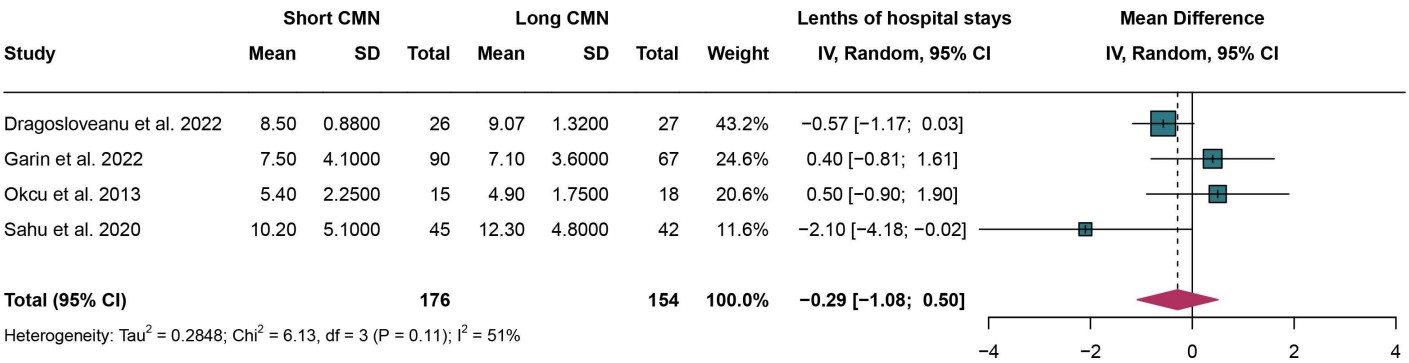

**Fig 12. Forest plot comparing lengths of hospital stays between short and long cephalomedullary nails groups.**

In 2021, Cinque et al.'s meta-analysis [28], which included 12 retrospective and prospective studies, found that short CMN were associated with significantly reduced estimated blood loss and shorter operative times compared to long CMN, with no significant difference in overall reoperation rates and peri-implant fractures. These findings align with our results;

however, Cinque et al.'s study lacked the higher evidence level provided by RCTs and did not analyze important functional outcomes such as Harris hip scores or stability indicators like tip-apex distance. Our study addresses these gaps by focusing on RCTs, providing a more robust and evidence-based comparison of the two CMN types.

In 2023, Rajnish et al. [14] conducted a meta-analysis that included 31 studies, most of which were retrospective. Their findings also highlighted the advantages of short CMN, including shorter duration of surgery, less intraoperative blood loss, and lower postoperative transfusion rates. However, they observed a significantly lower incidence of peri-implant fractures in the long CMN group, suggesting a potential benefit of long CMN in this aspect. In contrast, our study, which only included RCTs, did not find a significant difference between short and long CMN in the rate of peri-implant fractures. By limiting our analysis to RCTs, we provide more reliable evidence, but it should be noted that additional high-quality studies are necessary to fully understand the implications of CMN length on peri-implant fractures.

Short and long CMN each offer distinct advantages and disadvantages in the treatment of IFFs [29, 30]. Long CMN, which spans the entire length of the femur, can help reduce the risk of peri-implant fractures in elderly osteoporotic patients by avoiding stress concentration at the distal femoral shaft [14]. However, there is evidence suggesting that the insertion of a long CMN into the distal metaphysis may weaken the bone, increasing the risk of fractures [29,31]. Additionally, the insertion of long CMN is technically more demanding, requiring femoral canal reaming and freehand distal locking under fluoroscopic guidance, which can lead to iatrogenic fractures [32,33]. In contrast, short CMN are faster and technically simpler to insert, using a guide for distal locking, which reduces operative time and intraoperative complications [34,35]. Our results demonstrate that short CMN are associated with shorter duration of surgery, reduced tip-apex distance, and lower intraoperative blood loss compared to long CMN, suggesting that short CMN may be a better option in terms of efficacy. However, since the higher heterogeneity in some of our results and the relatively small number of included RCTs, further researches are needed to obtain more robust conclusions. In addition, while these differences are statistically significant, the clinical relevance of these findings depends on the context. Although shorter surgery duration and reduced blood loss may lead to faster recovery and lower risk of complications, it is important to consider whether these benefits are large enough to influence clinical practice. In many cases, the absolute difference in surgery duration (approximately 22 minutes) and blood loss (reduced by 136.7 mL) might be considered modest in the broader context of the patient's overall treatment and recovery [5,15]. Additionally, the choice between short and long CMN should not only rely on these intraoperative factors but also on other considerations such as fracture type, patient comorbidities, and cost [12,26]. The choice between short and long CMN in clinical practice should be tailored to the individual patient's condition, considering the specific advantages and potential risks of each option.

Some limitations of the current meta-analysis should be considered when interpreting the results. Firstly, the heterogeneity of the included studies may have impacted several outcomes, including the duration of surgery, lengths of hospital stays, and Harris hip scores. To assess the stability of these results, we conducted a leave-one-out sensitivity analysis, which revealed that only the Harris hip scores showed instability. This suggests that further research is needed to confirm the functional outcomes comparison between short and long CMN. Secondly, the relatively small number of randomized controlled trials (only 7 trials) included in this analysis, may introduce a small sample size bias. Consequently, well-designed randomized controlled trials are necessary to validate the findings of this meta-analysis and provide more robust evidence for clinical decision-making.

## 5. Conclusions

Short CMN are associated with shorter duration of surgery, reduced tip-apex distance, and lower intraoperative blood loss compared to long CMN for the fixation of IFFs. However, there were no significant differences in functional outcomes, overall complication rates, reoperation rates, mortality within one year, peri-implant fracture, or lengths of hospital stays. Future larger sample size randomized controlled trials are needed to validate the current conclusion.

## Supporting information

**S1 Checklist. PRISMA checklist: the PRISMA (Preferred Reporting Items for Systematic Reviews and Meta-Analyses) checklist used to ensure the transparency and completeness of the meta-analysis.**
(DOCX)

**S1 Raw data. Raw data used in this study, available for further analysis and verification.**
(XLSX)

**S1 File. Quality assessments: The quality assessments of the included studies.**
(DOCX)

**S2 File. Excluded and included studies Lists the studies that were included and excluded during the literature screening process.**
(XLSX)

**S3 File. Supplementary Materials: Search strategy utilized for literature retrieval and includes sensitivity analysis figures, showcasing the robustness of the findings.**
(DOCX)

## Acknowledgments

SZ helped in conceptualizing and designing the study, data collection and analysis, and writing the manuscript. Together, QG, KH, and HZ co-designed the study, as well as participated in data collection and analysis, and critically revised the content of the manuscript.

## Author contributions

**Conceptualization:** Haiqun Zhu.

**Data curation:** Shengquan Zhang, Qiaofeng Guo, Kai Huang, Haiqun Zhu.

**Formal analysis:** Qiaofeng Guo.

**Methodology:** Shengquan Zhang, Kai Huang.

**Software:** Shengquan Zhang.

**Validation:** Haiqun Zhu.

**Visualization:** Haiqun Zhu.

**Writing – original draft:** Shengquan Zhang, Qiaofeng Guo.

**Writing – review & editing:** Kai Huang, Haiqun Zhu.

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
