## [Decision Letter · Decision Letter 0]

12 Nov 2024

PONE-D-24-45695Short Versus Long Cephalomedullary Nails for Intertrochanteric Femur Fractures: A Meta-Analysis of Randomized Controlled TrialsPLOS ONE

Dear Dr. Zhu,

Thank you for submitting your manuscript to PLOS ONE. After careful consideration, we feel that it has merit but does not fully meet PLOS ONE’s publication criteria as it currently stands. Therefore, we invite you to submit a revised version of the manuscript that addresses the points raised during the review process.

The reviewers have indicated several remarks which can improve the paper. They are indicated below. Please consider them, and prepare the new version of the manuscript.

We look forward to receiving your revised manuscript.

Kind regards,

Pawel Klosowski, D.Sc.

Academic Editor

PLOS ONE

Journal Requirements:

5. If any table files for review show as item type ‘other’ please change to item type ‘Table’ as the reviewer does not have access to these ’other’ files.

6. We note that there is identifying data in the Supporting Information file <Supplementary Materials.docx>. Due to the inclusion of these potentially identifying data, we have removed this file from your file inventory. Prior to sharing human research participant data, authors should consult with an ethics committee to ensure data are shared in accordance with participant consent and all applicable local laws.

-Location data

Additional guidance on preparing raw data for publication can be found in our Data Policy (https://journals.plos.org/plosone/s/data-availability#loc-human-research-participant-data-and-other-sensitive-data ) and in the following article: http://www.bmj.com/content/340/bmj.c181.long .

Please remove or anonymize all personal information, ensure that the data shared are in accordance with participant consent, and re-upload a fully anonymized data set. Please note that spreadsheet columns with personal information must be removed and not hidden as all hidden columns will appear in the published file.

7. Please include captions for your Supporting Information files at the end of your manuscript, and update any in-text citations to match accordingly. Please see our Supporting Information guidelines for more information: http://journals.plos.org/plosone/s/supporting-information .

8. As required by our policy on Data Availability, please ensure your manuscript or supplementary information includes the following:

9. If any supporting files for review show as item type ‘other’ please change to item type ‘supporting info’ as the reviewer does not have access to these ’other’ files.

Reviewers' comments:

Reviewer's Responses to Questions

**Comments to the Author**

1. Is the manuscript technically sound, and do the data support the conclusions?

Reviewer #1: Yes

Reviewer #2: Yes

2. Has the statistical analysis been performed appropriately and rigorously? 

Reviewer #1: Yes

Reviewer #2: Yes

3. Have the authors made all data underlying the findings in their manuscript fully available?

Reviewer #1: Yes

Reviewer #2: Yes

4. Is the manuscript presented in an intelligible fashion and written in standard English?

Reviewer #1: Yes

Reviewer #2: Yes

5. Review Comments to the Author

Reviewer #1: Thank you for submitting this comprehensively written manuscript on short versus long cephalomedullary nails for intertrochanteric femur fractures. The methods decscribed are sound and results are decribed in a clear and concise fashion. It is however, not the first review on the topic, though it does include novel studies not included in previous research. The manuscript does also raise some questions and remarks that might warrant further revision or additional analysis before eventual publication.

Firstly, I do not quite understand why the authors have decided to make surgery time their primary outcome when they state in their aim to compare efficacy and safety, which are aguably much more important. Why not make functional outcomes or complication rates your primary outcome. Or describe in your introduction why you think surgery time is relevant as primary outcome for this analysis. In my opinion this would not be a deciding factor for superiority unless almost all other factors in this consideration were similar. Primary and secondary outcomes should be picked and described in order of importance (quality of life, mortality and functional outcomes -> complications and reoperation -> surgical outcomes and operation characteristics (like bloodloss or operation time). A significant difference does not mean that an outcomes should become the primary focus of a manuscript.

Several other comments and questions are listed below:

1) Line 13 (and throughout the manuscript) units (e.g. minutes or centimeters) are missing for many of the outcomes described.

2) Line 18: No need to mention all were P>0.05, this was already implied by no significant difference.

3) Line 19: It is not common to describe funnel plots results in the abstract, it is also unclear about which outcome the authors are writing here.

4)Line 37: Reference (6) is not about trochanteric fractures and therefore does not support the statement made. Please support this statement with adequate literature.

5) Line 37-40: Maybe this statement should be nuanced. Several reviews see only small differences between EMF and IMF, also in unstable fractures (AO type A2, not considering Type A3/reverse obliquity unstable fractures) See recent review by Zeelenberg et al. (PMID: 38175213) and Cochrane review by Lewis et al. (PMID: 35080771)

6) Line 41-42: The reasons for change in practise might be more nuance than only (suspected) difference in outcomes and might be highly surgeon or hospital related (Mellema et al, PMID: 33591214)

7) Methods: Did you also collect which type of trochanteric fracture was studied in the included studies? Could this have influence the outcomes of the meta-analysis? Should a subanalysis be conducted for various fracture types or is this not possible with the data available?

8) Line 100: Which complications were included in overall complication rate? What complications were included in this measure and were they different for the included studies?

9) Was tip-apex distance the only surgical outcome that was studied or did the studies also show data on other operation qualtity indicators.

10)Line 108: Were all studies sufficiently homogeneous in terms of included patient characteristics and/or fracture types to warrant usage of a fixed effects model? In some cases it may be better to use a random effects model for all outcomes if included study populations seem heterogeneous.

11) Line 121: Was it not possible to translate non-English articles to be able to include their data? If not, exclusion is a perfectly fine!

12) Line 137 (and throughout result section): I would not say quantitave results as you do not describe any qualitative results either. Just using the outcome compared as a title is sufficient.

13) Line 139: Please include the appropriate unit when describing outcomes (minutes or seconds or hours/mm cm meter)

14) Line 145: While the Eggers statistic did was non significant, about 50% of the included studies fell outside the funnel plot. Please comment on whether this may have had influence on the generalisability of these results.

15) Discussion: Please first list your main findings and then place them in a scientific context. The first paragraph is more or less a repetition of your introduction and does not explain the relevance of the findings of your analysis

16): Line 206-209: Please do not repeat extensive results (of many outcomes including CI's) in your discussion if not necessary, unless comparing results to those in other research.

17: Line 245: Despite the difference between groups, was the average tip apex distance not <25mm in both populations, suggesting adequate implant placement? Is this difference then a relevant outcome?

18) Discussion: Please comment on the clinical relevance of the differences you found. Is the magnitude of surgery duration or bloodloss great enough to change clinical practice? How do difference in costs influence these decisions?

19): Line 257: Was is the number of trials that was low or the number of included patients?

Kind regards

Reviewer #2: Line 21: “Future larger sample size randomized controlled trials are needed to validate the

current conclusion.“ Delete that from the abstract conclusions.

Line 26: “Intertrochanteric femur fractures (IFFs) are a common type of hip fracture, …“

No. They are not really hip fractures. They are a common type of proximal femur fractures.

Introduction: good.

Methods:

1. No study protocol registration in PROSPERO or elsewhere?

2. Why did not you search CNKI or other Chinese databases?= There is so many research going on in China.

3. Please give an exact defintion of short and long nails. How many cm?

Discussion

You should provide information what the best tip apex distance is in the literature. What does a reduced tip apex distance mean? Is it better or worse according to your findings under consideration oft he best tip apex distance discribed in the literature. Furthermore, report the overall mean tip apex distance of all short nail and all long nail studies. Are the values reported in the forest plot presented in mm? Then, you should discuss WHY there is a shorter tip apex distance in short nails, because the reason does not seem obvious. I cannot think of any reason why any operator would tend to proceed different according to shorter or longer nails in terms of the tip apex distance.

Conclusion

Line 261: Here you use “surgery times“ instead of the before mentioned “duration of surgery“. Unify your terms. The “reduced tip apex distance“ is an intersting finding, but it is hard to interprete because of the missiong information I mentioned before.

All in all, this is a valuable meta-analysis. Statistics and methods are fine. Good work. I am very interested in reading your changes.

6. PLOS authors have the option to publish the peer review history of their article (what does this mean? ). If published, this will include your full peer review and any attached files.

**Do you want your identity to be public for this peer review?** For information about this choice, including consent withdrawal, please see our Privacy Policy .

Reviewer #1: No

Reviewer #2: No

---

## [Author Response · Author response to Decision Letter 1]

13 Jan 2025

Reviewer #1

Overall comment:

Thank you for submitting this comprehensively written manuscript on short versus long cephalomedullary nails for intertrochanteric femur fractures. The methods described are sound and results are described in a clear and concise fashion. It is however, not the first review on the topic, though it does include novel studies not included in previous research. The manuscript does also raise some questions and remarks that might warrant further revision or additional analysis before eventual publication.

Firstly, I do not quite understand why the authors have decided to make surgery time their primary outcome when they state in their aim to compare efficacy and safety, which are arguably much more important. Why not make functional outcomes or complication rates your primary outcome. Or describe in your introduction why you think surgery time is relevant as primary outcome for this analysis. In my opinion this would not be a deciding factor for superiority unless almost all other factors in this consideration were similar. Primary and secondary outcomes should be picked and described in order of importance (quality of life, mortality and functional outcomes -> complications and reoperation -> surgical outcomes and operation characteristics (like blood loss or operation time). A significant difference does not mean that an outcome should become the primary focus of a manuscript.

Response:

Thank you for your feedback. Based on your suggestion, we have revised the manuscript to focus on functional outcomes and mortality as primary outcome measures, reflecting the importance of efficacy and safety in clinical practice. The duration of surgery has now been moved to a secondary outcome.

The revised section now reads:

"The primary outcome measures were functional outcomes, specifically the Harris Hip Score, and mortality within 1 year. Secondary outcomes included complication rates (any adverse events occurring during or after surgery), reoperation rates, the duration of surgery (time from the initial incision to final wound closure), tip-apex distance, intraoperative blood loss, and length of hospital stay."

Additionally, we have updated the order of figures and the presentation of results to reflect this revised prioritization.

Comment 1:

Line 13 (and throughout the manuscript) units (e.g. minutes or centimeters) are missing for many of the outcomes described.

Response:

Thank you for your comment. We agree with your observation and have added the appropriate units (minutes for surgery time, centimeters for tip-apex distance, and milliliters for intraoperative blood loss) throughout the manuscript to ensure clarity and precision in the reported outcomes. The updated paragraph is as follows:

Results: A total of 7 studies with 658 patients were included in this analysis. Durations of surgery were significantly lower in the short CMN group compared to the long CMN group (MD: -21.83 minutes, 95% CI: -27.54 minutes, -16.13 minutes). Furthermore, tip-apex distance was significantly lower in the short CMN group compared to the long CMN group (MD: -0.47 cm, 95% CI: -0.63 cm, -0.31 cm), along with a lower intraoperative blood loss (MD: -136.70 mL, 95% CI: -139.06 mL, -134.34 mL).

Comment 2:

Line 18: No need to mention all were P>0.05, this was already implied by no significant difference.

Response:

Thank you for your suggestion. We have revised the manuscript by removing the reference to P>0.05

Comment 3:

Line 19: It is not common to describe funnel plots results in the abstract, it is also unclear about which outcome the authors are writing here.

Response:

Thank you for your valuable feedback. Regarding your comment on line 19 about the description of funnel plot results, we have removed the related statement as per your suggestion.

Comment 4:

Line 37: Reference (6) is not about trochanteric fractures and therefore does not support the statement made. Please support this statement with adequate literature.

Response:

Thank you for pointing this out. We have replaced reference (6) with a more appropriate source to support the statement. The new reference is:

Reindl, Rudolf et al. “Intramedullary Versus Extramedullary Fixation for Unstable Intertrochanteric Fractures: A Prospective Randomized Controlled Trial.” The Journal of Bone and Joint Surgery, American Volume, vol. 97, no. 23 (2015): 1905-12. doi:10.2106/JBJS.N.01007.

We believe this provides a stronger basis for the statement made.

Comment 5:

Line 37-40: Maybe this statement should be nuanced. Several reviews see only small differences between EMF and IMF, also in unstable fractures (AO type A2, not considering Type A3/reverse obliquity unstable fractures) See recent review by Zeelenberg et al. (PMID: 38175213) and Cochrane review by Lewis et al. (PMID: 35080771)

Response:

Thank you for your valuable feedback. We have revised the relevant section based on your suggestion and cited the proper refences. The updated version is as follows:

"However, several recent reviews suggest that the differences between CMN and DHS are small, particularly in unstable fractures such as AO type A2 fractures, although not including reverse obliquity fractures (8, 9). Despite this, CMN are still increasingly preferred over DHS for the treatment of unstable intertrochanteric fractures, partly due to their ability to provide superior stability in certain fracture patterns (10)."

Comment 6:

Line 41-42: The reasons for change in practice might be more nuance than only (suspected) difference in outcomes and might be highly surgeon or hospital related (Mellema et al, PMID: 33591214)

Response:

Thank you for your insightful comment. We have revised the relevant section to reflect the nuance in the reasons for the shift in practice, as suggested. The updated version is as follows:

“Despite this, CMN are still increasingly preferred over DHS for the treatment of unstable intertrochanteric fractures, partly due to their ability to provide superior stability in certain fracture patterns(10). This change in practice might also be influenced by factors such as surgeon preference, hospital protocols, and regional variations(11).”

Comment 7:

Methods: Did you collect which type of trochanteric fracture was studied in the included studies? Could this influence the outcomes of the meta-analysis? Should a subanalysis be conducted for various fracture types?

Response:

Thank you for your valuable comment. Due to the complexity of the data, we were unable to conduct a subgroup analysis based on different types of trochanteric fractures. However, in response to your suggestion, we have added a new column titled "AO/OTA Fracture" in Table 1. This column includes the distribution of the AO/OTA fracture classifications for both the Short CMN and Long CMN groups. The updated content can be found in the revised Table 1.

Comment 8:

Line 100: Which complications were included in overall complication rate? What complications were included in this measure and were they different for the included studies?

Response:

We have added more detailed information regarding the complications included in the overall complication rate in the results section of the manuscript.

The complications reported in the studies were as follows: Dragosloveanu et al. (2022) identified screw cut-out and local postoperative hematoma; Garín et al. (2022) noted mechanical issues such as blade lateral migration, cut-through, malrotation, pin fatigue/fracture, and non-union, along with infections and other general complications; Galanopoulos et al. (2018) reported the z-effect phenomenon and periprosthetic fractures; Okcu et al. (2013) observed superficial and deep surgical site infections; and Shannon et al. (2019) mainly reported screw cut-out and deep surgical site infection.

Comment 9:

Was tip-apex distance the only surgical outcome that was studied or did the studies also show data on other operation quality indicators?

Response:

Thank you for your comment. Tip-apex distance was not the only surgical outcome studied. We also evaluated several other surgical quality outcomes(such as Harris Hip Score).

However, tip-apex distance is an important stability indicator, and to the best of our knowledge, it has not been included in previous meta-analyses. In our study, we introduced this measure as a new surgical outcome to assess the differences between the Short CMN and Long CMN groups.

Comment 10:

Line 108: Were all studies sufficiently homogeneous in terms of included patient characteristics and/or fracture types to warrant usage of a fixed effects model?

Response:

Thank you for your comment. In response to your concern, we reviewed the fracture types across the included studies. Upon re-examination, we found that the majority of patients were categorized into 31-A2 and 31-A3 fractures(We have added this information in Table 1). However, we acknowledge that the fracture types were not entirely homogeneous across all studies, which is a limitation of any meta-analysis.

Regarding the choice of statistical model, we selected the fixed-effects model based primarily on the statistical homogeneity of the data. Specifically, when the Cochran’s Q test yielded a P-value less than 0.10 and the I2 statistic was less than 50%, we opted for the fixed-effects model[1]. This approach ensures that the model used was appropriate based on the observed data heterogeneity.

We hope this clarifies the rationale behind our choice of statistical model.

[1] Yang K, Kwan H Y, Yu Z, et al. Model selection between the fixed-effects model and the random-effects model in meta-analysis[J]. Statistics and its Interface, 2020, 13(4): 501-510.

Comment 11:

Line 121: Was it not possible to translate non-English articles to be able to include their data?

Response:

Thank you for your comment. Regarding your question about non-English articles, we initially set clear inclusion and exclusion criteria that specifically excluded studies published in languages other than English. This decision was made for several reasons: First, non-English literature can be difficult for readers to access and interpret, which could limit the broader impact and utility of the findings. Second, many studies published in languages such as Spanish or Chinese may have lower methodological rigor or may not meet the same quality standards as those published in English.

As such, we only included studies that had full-text articles available in English. We have emphasized this in our inclusion and exclusion criteria: "Additionally, studies published in languages other than English were excluded due to accessibility issues for readers and concerns regarding the consistency of study quality across different languages."

Comment 12:

Line 137 (and throughout result section): I would not say quantitative results as you do not describe any qualitative results either. Just using the outcome compared as a title is sufficient.

Response:

Thank you for your comment. We will revise the manuscript accordingly. Instead of using the term "quantitative results," we will refer to the outcomes by their specific titles, such as "Harris Hip Score," "mortality within 1 year," "complication rates," etc., to maintain clarity and precision in describing the findings. We have revised in the manuscript.

Comment 13:

Line 139: Please include the appropriate unit when describing outcomes (minutes or seconds or hours/mm cm meter)

Response:

Thank you for your comment. We have added the appropriate units to the outcomes where applicable. This revision has been made throughout the results section.

Comment 14:

Line 145: While the Eggers statistic did was non-significant, about 50% of the included studies fell outside the funnel plot. Please comment on whether this may have had influence on the generalizability of these results.

Response:

Thank you for your comment. We have revised the paragraph to address your concern. Here is the updated version:

“Additionally, funnel plot analysis and Egger's test showed no significant publication bias (P = 0.07) (Figure 8). However, it is worth noting that approximately 50% of the included studies fell outside the funnel plot, which may raise concerns about the potential influence on the generalizability of these results.”

Comment 15:

Discussion: Please first list your main findings and then place them in a scientific context. The first paragraph is more or less a repetition of your introduction and does not explain the relevance of the findings of your analysis.

Response:

Thank you for your valuable feedback. We have revised the first paragraph of the discussion section as suggested. The updated version now clearly presents our main findings and provides their scientific context. Below is the modified text:

Revised First Paragraph:

In the treatment of IFFs, intramedullary nails are a widely accepted and effective option, with both long and short CMN offering distinct advantages(27). Our meta-analysis demonstrated that short CMN are associated with significantly shorter durations of surgery, lower tip-apex distance, and reduced intraoperative blood loss compared to long CMN in the fixation of IFFs. These findings may be attributed to the technical differences between the two types of nails. Short CMN are quicker to insert due to their reduced length, which requires less reaming of the femoral canal and simpler distal locking procedures(15, 16, 18). The shorter length also minimizes the distance the implant must traverse, leading to a reduced tip-apex distance, which is a crucial factor in achieving optimal screw placement within the femoral head(18). Furthermore, the less invasive nature of the procedure with short CMN results in lower intraoperative blood loss, likely due to decreased soft tissue and bone disruption during insertion(15, 16, 18).

We hope this revised version addresses your concerns and improves the clarity of our findings.

Comment 16:

Line 206-209: Please do not repeat extensive results (of many outcomes including CI's) in your discussion if not necessary, unless comparing results to those in other research.

Response:

Thank you for your comment. We have revised the discussion section by removing the extensive results and confidence intervals, as suggested. Only key points are now presented.

Comment 17:

Line 245: Despite the difference between groups, was the average tip apex distance not <25mm in both populations, suggesting adequate implant placement? Is this difference then a relevant outcome?

Response:

Thank you for your valuable comment regarding the tip-apex distance(TAD). We agree that both groups in our meta-analysis had an average TAD<25mm, which generally suggests adequate implant placement and is associated with favorable clinical outcomes. However, the statistically significant difference in TAD between the short and long CMN groups still warrants consideration.

While the absolute difference in TAD between the two groups may be small, it is important to note that even minor variations in TAD can have clinical relevance, particularly in complex fractures. In cases where the bone quality is compromised or the fracture pattern is unstable, small differences in screw placement may influence the overall stability of the implant and potentially affect long-term outcomes, such as the risk of screw cut-out or implant failure[1]. Additionally, achieving an optimal TAD is crucial for minimizing complications, even if the measurements are within an acceptable range[2]. Thus, we believe that this difference, albeit small, may still prov

---

## [Decision Letter · Decision Letter 1]

22 Jan 2025

PONE-D-24-45695R1Short Versus Long Cephalomedullary Nails for Intertrochanteric Femur Fractures: A Meta-Analysis of Randomized Controlled TrialsPLOS ONE

Dear Dr. Zhu,

Thank you for submitting your manuscript to PLOS ONE. After careful consideration, we feel that it has merit but does not fully meet PLOS ONE’s publication criteria as it currently stands. Therefore, we invite you to submit a revised version of the manuscript that addresses the points raised during the review process.

Only Reviewer 1 has submitted the opinion on the current version of the paper. As only minor revisions are required I decided to stop the reviewing process in that moment. Please try to introduce the current Reviewer requirements and submit the final version of the paper.

We look forward to receiving your revised manuscript.

Kind regards,

Pawel Klosowski, D.Sc.

Academic Editor

PLOS ONE

Journal Requirements:

Reviewers' comments:

Reviewer's Responses to Questions

**Comments to the Author**

1. If the authors have adequately addressed your comments raised in a previous round of review and you feel that this manuscript is now acceptable for publication, you may indicate that here to bypass the “Comments to the Author” section, enter your conflict of interest statement in the “Confidential to Editor” section, and submit your "Accept" recommendation.

Reviewer #1: (No Response)

2. Is the manuscript technically sound, and do the data support the conclusions?

Reviewer #1: Yes

3. Has the statistical analysis been performed appropriately and rigorously? 

Reviewer #1: Yes

4. Have the authors made all data underlying the findings in their manuscript fully available?

Reviewer #1: Yes

5. Is the manuscript presented in an intelligible fashion and written in standard English?

Reviewer #1: Yes

6. Review Comments to the Author

Reviewer #1: Thank you for your detailed answers to my comments and the corresponding changes to the manuscript. It has improved considerably in its clarity and discussion of its findings. I think the current manuscripts findings can be a good addition to the current literature on the topic. I have but 2 further suggestions:

1. Maybe the authors could also follow the same order of results in the manuscript abstract as in the methods/results. Now the result section of the abstract still starts with the significant results and not with function or complications.

2. Please include no differences in functional outcomes, overall complications, and reoperation in the conclusion of your abstract and conclusion section. This information could also be given a more prominent place in the first paragraph of the discussion section. It is important to keep in mind that while TAD, operation time, and bloodloss were significantly different, the absolute differences were small and the clinical relevance of these differences are debatable and likely small. The two groups showed no differences in other (arguably more important) outcomes. Your discussion/conclusions should reflect this.

Kind regards

7. PLOS authors have the option to publish the peer review history of their article (what does this mean? ). If published, this will include your full peer review and any attached files.

**Do you want your identity to be public for this peer review?** For information about this choice, including consent withdrawal, please see our Privacy Policy .

Reviewer #1: No

---

## [Author Response · Author response to Decision Letter 2]

22 Jan 2025

Reviewer #1

Comment 1:

Maybe the authors could also follow the same order of results in the manuscript abstract as in the methods/results. Now the result section of the abstract still starts with the significant results and not with function or complications.

Response:

Thank you for this helpful suggestion. We have revised the results section of the abstract to follow the same order as the results section in the manuscript (3.3–3.11). Specifically, we now present the findings related to Harris hip score, 1-year mortality, overall complications, and reoperation rates before describing the statistically significant differences in operation duration, intraoperative blood loss, and tip-apex distance. The revised abstract result section is as follows:

"A total of 7 studies with 658 patients were included in this analysis. There was no significant difference between the short CMN group and the long CMN group in Harris hip score, mortality within 1-year, overall complication rates, or reoperation rates. However, durations of surgery were significantly lower in the short CMN group compared to the long CMN group (MD: -21.83 minutes, 95% CI: -27.54 minutes, -16.13 minutes), along with significantly lower intraoperative blood loss (MD: -136.70 mL, 95% CI: -139.06 mL, -134.34 mL) and tip-apex distance (MD: -0.47 cm, 95% CI: -0.63 cm, -0.31 cm). There was also no significant difference in peri-implant fracture or lengths of hospital stays."

We hope this revised structure addresses your concern and improves the clarity of the abstract.

Comment 2:

Please include no differences in functional outcomes, overall complications, and reoperation in the conclusion of your abstract and conclusion section. This information could also be given a more prominent place in the first paragraph of the discussion section. It is important to keep in mind that while TAD, operation time, and bloodloss were significantly different, the absolute differences were small and the clinical relevance of these differences are debatable and likely small. The two groups showed no differences in other (arguably more important) outcomes. Your discussion/conclusions should reflect this.

Response:

Thank you for this insightful suggestion. We have made the following revisions to address your comment:

Conclusions:

We have revised the conclusion section to include the lack of significant differences in functional outcomes, overall complication rates, reoperation rates, and mortality within one year. The updated conclusion now reads as follows:

"Short CMN are associated with shorter duration of surgery, reduced tip-apex distance, and lower intraoperative blood loss compared to long CMN for the fixation of IFFs. However, there were no significant differences in functional outcomes, overall complication rates, reoperation rates, mortality within one year, peri-implant fracture, or lengths of hospital stays."

Discussion:

To reflect the limited clinical relevance of the statistically significant differences, we added the following sentence to the first paragraph of the discussion section:

"These differences were statistically significant but small in absolute terms."

---

## [Decision Letter · Decision Letter 2]

7 Feb 2025

Short Versus Long Cephalomedullary Nails for Intertrochanteric Femur Fractures: A Meta-Analysis of Randomized Controlled Trials

PONE-D-24-45695R2

Dear Dr. Zhu,

We’re pleased to inform you that your manuscript has been judged scientifically suitable for publication and will be formally accepted for publication once it meets all outstanding technical requirements.

Kind regards,

Pawel Klosowski, D.Sc.

Academic Editor

PLOS ONE

Additional Editor Comments (optional):

Reviewers' comments:

Reviewer's Responses to Questions

**Comments to the Author**

1. If the authors have adequately addressed your comments raised in a previous round of review and you feel that this manuscript is now acceptable for publication, you may indicate that here to bypass the “Comments to the Author” section, enter your conflict of interest statement in the “Confidential to Editor” section, and submit your "Accept" recommendation.

Reviewer #1: All comments have been addressed

2. Is the manuscript technically sound, and do the data support the conclusions?

Reviewer #1: (No Response)

3. Has the statistical analysis been performed appropriately and rigorously? 

Reviewer #1: (No Response)

4. Have the authors made all data underlying the findings in their manuscript fully available?

Reviewer #1: (No Response)

5. Is the manuscript presented in an intelligible fashion and written in standard English?

Reviewer #1: (No Response)

6. Review Comments to the Author

Reviewer #1: Thank you for your final adjustments. If the editor agrees, I find the manuscript suitable for publication

7. PLOS authors have the option to publish the peer review history of their article (what does this mean? ). If published, this will include your full peer review and any attached files.

**Do you want your identity to be public for this peer review?** For information about this choice, including consent withdrawal, please see our Privacy Policy .

Reviewer #1: **Yes: ** Dennis Den Hartog

---

## [Editor Report · Acceptance letter]

PONE-D-24-45695R2

PLOS ONE

Dear Dr. Zhu,

I'm pleased to inform you that your manuscript has been deemed suitable for publication in PLOS ONE. Congratulations! Your manuscript is now being handed over to our production team.

Kind regards,

on behalf of

Prof. Pawel Klosowski

Academic Editor

PLOS ONE